# Epidemiology and Management of Cerebral Venous Thrombosis during the COVID-19 Pandemic

**DOI:** 10.3390/life12081105

**Published:** 2022-07-22

**Authors:** Natalia Novaes, Raphaël Sadik, Jean-Claude Sadik, Michaël Obadia

**Affiliations:** 1Hôpital Fondation Adolphe de Rothschild, Stroke Unit, 75019 Paris, France; natalianovaes.92@gmail.com; 2Hospital Riviera-Chablais, Unit of Geriatrics Rehabilitation, 1800 Vevey, Switzerland; raphael.sadik@hopitalrivierachablais.ch; 3Hôpital Fondation Adolphe de Rothschild, Radiology, 75019 Paris, France; jcsadik@for.paris

**Keywords:** cerebral venous thrombosis, COVID-19, adenovirus vaccine, vaccine-induced immune thrombotic thrombocytopenia (VITT)

## Abstract

Cerebral venous thrombosis (CVT) is a rare type of stroke that may cause an intracranial hypertension syndrome as well as focal neurological deficits due to venous infarcts. MRI with venography is the method of choice for diagnosis, and treatment with anticoagulants should be promptly started. CVT incidence has increased in COVID-19-infected patients due to a hypercoagulability state and endothelial inflammation. CVT following COVID-19 vaccination could be related to vaccine-induced immune thrombotic thrombocytopenia (VITT), a rare but severe complication that should be promptly identified because of its high mortality rate. Platelet count, D-dimer and PF4 antibodies should be dosed. Treatment with non-heparin anticoagulants and immunoglobulin could improve recuperation. Development of headache associated with seizures, impaired consciousness or focal signs should raise immediate suspicion of CVT. In patients who received a COVID-19 adenovirus-vector vaccine presenting thromboembolic events, VITT should be suspected and rapidly treated. Nevertheless, vaccination benefits clearly outweigh risks and should be continued.

## 1. Introduction

Cerebral venous thrombosis (CVT) is an uncommon type of stroke with an incidence around 3–4 cases per million adults, with a predominance in Caucasian females [1]. It may be associated with five major clinical syndromes:Increased intracranial pressure with newly onset headache in 70 to 90% of patients [2], and less frequently papilledema and palsy of the abducent nerve;Seizures, which may affect as much as 34% of patients [3];Focal neurological deficits [4];Encephalopathy or coma, usually in the context of multiple sinus or deep venous occlusions;Specific regional syndromes, including periorbital pain, ocular chemosis, palsies of oculomotor cranial nerves and the ophthalmic division of the fifth cranial nerve (in the case of cavernous sinus thrombosis).

The intracranial hypertension is due to increased venular and capillary pressure as well as decreased cerebrospinal fluid absorption.

There are several risk factors associated with CVT, such as pregnancy, postpartum and the use of oral contraceptives [5]. Other conditions such as hypercoagulability states in malignancy or genetic predisposition (mutation of Factor V Leiden, deficiency in proteins C or S, prothrombin mutation, hyperhomocysteinemia) may also be associated with this disease. Cancer treatment by tamoxifen, cisplatin or other chemotherapies is another rare cause of CVT. Specific neurological diseases such as cranial trauma, neurological intervention and bacterial meningitis are other possible causes of this disorder. In children particularly, acute head and neck infections (such as mastoiditis) are the primary causes of CVT. Finally, Behçet’s disease and Systemic Lupus Erythematosus are the main systemic diseases associated with CVT [4].

By May 2022, COVID-19 had infected 513 million people worldwide and had caused over 6.2 million deaths [6]. Part of its morbidity and mortality has been related to the potential increase in thromboembolic events, such as CVT, described in several case reports [7]. A retrospective multicenter cohort study reported an incidence of 8.8 per 10,000 [8], versus 3–4 per million in previous publications before the pandemics [1,9,10,11,12]. Thus, its prevalence is higher in COVID-19-positive patients than in the general population, with a predominance of males (66%) and a mean age of 48 years. This was also suggested by other epidemiological studies that reported no predominance of young female patients, as is the case with classic CVT [13].

The aims of this study, therefore, are to report cases of patients with CVT related to COVID-19 who were hospitalized in the Neurovascular Unit of the Hôpital Fondation Adolphe de Rothschild in Paris and, afterwards, to review the literature concerning the management of CVT in the COVID-19 pandemic context.

## 2. Materials and Methods

A retrospective analysis of medical charts of the Neurovascular Unit of Hôpital Fondation Adolphe de Rothschild was performed. The number of hospitalizations due to cerebral venous thrombosis from 2018 to 2020 was compared with the period of 2020 to 2021. Cases in which COVID-19 infection was detected in relation to CVT were described.

Descriptive statistics were performed using JASP (JASP Team (2022), Version 0.16.3) (computer software), Eric-Jan Wagenmakers, University of Amsterdam. Amsterdam, The Netherlands.

A literature review was developed on the Pubmed database searching for the keywords “cerebral venous thrombosis”, “COVID-19”, “COVID vaccine”, and “vaccine-induced thrombotic thrombocytopenia”.

Patients and/or patients’ families consented to the publication of images and data.

## 3. Results

During the period of 2018–2019, 39 individuals with CVT were hospitalized, while 47 patients were admitted during 2020–2021 (a 1.2-fold increase). However, COVID-19 PCR was positive in only 5 patients out of those 47 (10.6% of cases). Only one case of CVT was possibly related to a COVID-19 vaccine (Johnson & Johnson).

Table 1 shows the clinical and epidemiological data of patients presenting COVID-19 and CVT. All the patients were women, aged from 31 to 61 (mean 45.8, SD 13.9). Mean time from COVID-19 diagnosis to CVT was 18.6 days (SD 12.4), and the mean time from first CVT symptoms to diagnosis was 18.4 days (SD 11.8).

One patient had a history of breast cancer 5 years before evaluation. It was in remission, but still being treated by tamoxifen. Another individual was under oral contraception with an estrogen–progestogen combined pill for the treatment of menorrhagia and had an iron-deficiency anemia with a hemoglobin level of 7 g/dL. One patient was obese, with a Body Mass Index (BMI) of 34 kg/m^2^ and had a previous history of five miscarriages, with anti-phospholipid syndrome ruled out.

Regarding neuroimaging features, three of our patients had transverse and sigmoid sinus thrombosis, one of which was also associated with a hemorrhagic venous infarct (Figure 1); one patient had superior sagittal and right transverse sinus thrombosis, and another had multiple sinuses, including the deep venous system. The latter also had venous infarct with hemorrhagic transformation (Figure 2) and underwent craniectomy and thrombectomy, but unfortunately died due to multiple neurological and respiratory complications.

A 70-year-old male patient was hospitalized for presenting CVT with involvement of the superior longitudinal sinus and the left transverse sinus (Figure 3). Such events took place three weeks after receiving a Johnson & Johnson vaccine. He had no other risk factors for thrombosis and presented a normal platelet count on diagnosis (246 G/L). He was treated with low-molecular-weight heparin (LMWH), later switched to Dabigatran, and had a favorable outcome.

## 4. Discussion

CVT is a rare but potentially severe neurological disorder that may have a wide spectrum of clinical presentations.

The method of brain Magnetic Resonance Imaging (MRI) with contrast-enhanced venography is the one most commonly indicated to diagnose CVT. The diversity of radiological signs depends on the delay between symptom onset and MRI, but the typical finding consists of the visualization of the thrombus, a contrast-filling defect (empty delta sign), or the lack of flow signal. Venous strokes appear as an edematous region with mixed infarct, with possible hemorrhage in atypical locations, not compatible with arterial territories. If a venous MRI is not available, or for any reason contraindicated, a Computerized Tomography (CT) scan with venography is an alternative diagnostic modality.

Concerning treatment, it should consist of anticoagulation by low-molecular-weight heparin, warfarin or with Direct Oral Anticoagulants (DOACs). The RESPECT-CVT trial, which compared dabigatran versus warfarin in 120 randomized patients, reported no differences in the safety and efficacy between the groups [14]. Yaghi and colleagues [15] performed a retrospective multicentric international observational cohort including over 1000 patients. Real-world cases of CVT that were treated by either warfarin or DOACs were compared. Patients treated with DOACs presented similar rates of recanalization, death and recurrence, but with lower risk of major hemorrhage, than patients treated with warfarin [15]. Patients with a clear indication for warfarin (such as presence of anti-phospholipid antibody syndrome) were excluded.

A randomized trial compared Rivaroxaban versus warfarin or heparin in 114 children with CVT and found decreased bleeding risk with similar low recurrence rates and reduced thrombotic burden [16].

Endovascular treatment is mostly discussed in cases of thrombosis of multiple sinuses associated with severe intracranial hypertension and clinical deterioration. The American Heart Association guidelines recommend endovascular thrombolysis or thrombectomy in refractory patients who deteriorate despite anticoagulant treatment [17]. Systematic reviews published in 2015 and 2017 suggested that mechanical thrombectomy is safe [18] and effective in salvage therapy for refractory CVT [19]. A small case series of seven patients who presented clinical deterioration despite the best medical treatment reported good outcomes after such intervention [20]. A retrospective study including 30 patients with neurological deterioration or refractory seizures reported that combined endovascular mechanical thrombectomy and on-site chemical thrombolysis were reasonably safe and might be considered as an option for severe cases that are unresponsive to anticoagulation, especially for thrombosis of the superior sagittal sinus [21].

However, it is worth mentioning that a multicenter, randomized, open-label study (the TO-ACT study) that included 67 patients with CVT and at least one risk factor for poor outcome (coma, alteration in mental status, intracerebral hemorrhage or deep vein thrombosis) showed no difference in disability at 12 months between patients treated by a standard medical treatment versus those who underwent endovascular treatment [22]. The study was prematurely ended due to futility.

Decompressive craniectomy is a controversial procedure that can be performed in life-threatening situations. Two retrospective studies of medical records compared patients with malignant CVT treated by craniectomy versus medical treatment alone, and the results showed that craniectomy decreased mortality and improved functional outcomes [23,24]. A prospective small case series also suggested that this intervention can be lifesaving, with good clinical outcomes [25].

Antiepileptic drugs should not be used preventively, and corticoids have no benefit. Acetazolamide could eventually be useful in refractory intracranial hypertension [17].

Considering prognosis, it is largely variable depending on the site of thrombosis, the size of thrombus and also on the delay before clinical diagnosis and the start of treatment. The overall mortality is around 6% while circa 10% have permanent disability after one year of follow-up [26].

### 4.1. CVT Related to COVID-19 Infection: A Literature Review

The clinical presentation of CVT related to COVID-19 does not seem to be different from CVT which is not COVID-19-related: cephalalgia, seizures, encephalopathy, abducent palsies, papilledema and focal deficits such as hemiparesis or aphasia. In a multicenter study, headache was present in 50% of the cases, decreased level of consciousness in 12.5% and neurological focal deficits in 25% of patients. In another multicentric retrospective cohort, headache was present in 85% of cases. Focal symptoms such as hemiparesis or seizures were present in 42% of cases, and cortical signs such as aphasia or hemianopsia were present in 25% [8]. Possible complications, as in non-COVID-19-related CVT, were intracranial bleeding, seizures and decreased consciousness with intubation. A multicentric study reported intraparenchymal hemorrhage and subarachnoid hemorrhage in up to 25% of patients [27].

One study focused on neuro-ophthalmological complications of CVT related to COVID-19. The intracranial hypertension caused by CVT can lead to papilledema and direct ischemic injury of visual pathways and also to ocular motility impairment due to injury of oculomotor and abducent nerves with diplopia. For patients with papilledema and visual impairment, a lumbar puncture can be performed to rapidly lower intracranial pressure. Acetazolamide can also be used in this context. Severe cases should undergo cerebrospinal fluid shunt or optic nerve fenestration [28].

Thromboembolic events related to COVID-19 are frequent and can appear in later stages of infection and even after patients no longer have respiratory symptoms [7]. One study from Wuhan reported an incidence of deep venous thrombosis of 21% amongst COVID-19 patients [29]. Another paper reported that COVID-19 patients present higher levels of D-dimer and fibrinogen [30]. One review showed that 75% of patients had elevated D-dimers and 50% had increased CRP [13]. The presence of antiphospholipid antibodies was also described [31]. Also with regard to biological data, lumbar puncture was often performed, and was usually non-collaborative, showing mild protein elevation in cerebrospinal fluid [13].

It is worthy of mention that even “Long COVID-19 syndrome”—which refers to patients who present long-lasting inflammatory alterations and prolonged persistence of symptoms—can be associated with an increase in prothrombotic state. Some of the patients present persistent high levels of D-dimers with an increased risk of thromboembolic complications [32].

The literature reports 3 to 4 days as the mean time from neurological symptoms onset to CVT diagnosis and 7 to 11 days as the mean time from COVID-19 symptoms onset to CVT diagnosis [8,13,27,33]. CVT also occurred in patients with mild respiratory symptoms, and no direct association between systemic severity and risk of CVT was found [27].

In a multicentric study, Abdalkader and colleagues [22] reported hypertension and diabetes as the most common comorbidities among patients. Risk factors such as obesity, intake of estrogen contraceptives and antiphospholipid syndrome were also present.

Regarding neuroimaging features, the transverse sinuses and the superior sagittal were most commonly affected; one patient also had deep venous thrombosis [27]. According to the International Study on Cerebral Vein and Dural Sinus Thrombosis, thrombosis of the deep venous system is an independent predictor of death [34].

Another case series showed that half of the patients presented multiple sinuses implicated [13]. One review suggested that there is a trend for COVID-19-related CVT patients to have hemorrhagic infarcts more often in the first admission neuroimage, although data were not statistically confirmed [35].

It has been well established in the literature that infections, in general, increase the risk of CVT, especially those related to the cranium, such as sinusitis, meningitis, mastoiditis, otitis and tonsillitis [28]. Viral infections were also associated with thromboembolic risk, such as Human Immunodeficiency Virus (HIV), Ebola, cytomegalovirus and varicella zoster [36]. These findings reinforce the link between infection and CVT.

The mechanisms which lead to the increase in thromboembolic events vary and include endothelial disfunction, cytokine storm, an increase in prothrombotic markers such as D-dimers and fibrinogen, as well as the rise of pro-inflammatory molecules such as interleukin-6 and C Reactive Protein (CRP), which would result in a hypercoagulability state [7]. The sepsis induced by the infection may also contribute to thromboembolic events [33].

COVID-19’s interaction with angiotensin-converting enzyme (ACE) receptors may be the leading cause of endothelial damage, being associated with the coagulopathy caused by the increase in prothrombotic molecules such as D-dimers and fibrinogen, the dysregulation of inflammatory cascades as well as platelet dysfunction. The latter is characterized by increased platelet adherence and aggregation [37], which in turn would lead to an increase in thromboembolic events, which might be as severe as intravascular coagulation [35]. The increased viscosity can also potentialize endothelial damage, resulting in impaired microcirculation. The immune response to the virus can trigger complement activation, which culminates in prostaglandin and proinflammatory cytokines—the so-called cytokine storm—that also leads to altered coagulation. Antiphospholipid antibodies could also play a role in COVID-19-related CVT, as they were found in a few critical patients with COVID-19 and stroke [31].

Nevertheless, hypoxia, which is frequently present in COVID-19 patients, especially in the intensive care unit setting, is also associated with increased blood viscosity, the activation of genes related to hypoxia that interfere with coagulation and alterations in fibrinolysis [33].

Standard CVT treatment, with LMWH used in most cases, was associated with the treatment of complications, such as antiepileptic drugs when seizures were present or detected in the EEG of comatose patients. Very few patients were treated by endovascular thrombectomy or thrombolysis. A multicentric study already mentioned above recommends intravenous recombinant tissue plasminogen activator (rt-PA) and endovascular treatment as a last resort for refractory cases [27,38]. One multicentric retrospective cohort described two patients treated by endovascular thrombolysis, and craniectomy was also performed in two patients [8].

Concerning prognosis, one Asiatic review found a mortality rate in CVT related to COVID-19 as high as 45.5% [13]. CVT mortality in the non-COVID population is estimated to be below 10% [39]. In the same retrospective cohort mentioned above, mortality was 25%, 50% were transferred to a rehabilitation facility and 25% were discharged home [8].

### 4.2. CVT Associated with COVID-19 Vaccines

Not only COVID-19 infection but also COVID-19 vaccines were associated with thromboembolic episodes in many reports and reviews. To this date, 11.5 billion doses of COVID-19 vaccines have been administered worldwide [6], which has dramatically decreased morbidity and mortality related to the disease.

The rapid development of the vaccines was due, partially, to previous research on the technology of adenoviral vectors, which, along with m-RNA technology, accounts for the majority of vaccines that were administered during the pandemic period.

One of the most severe adverse effects attributed to the adenovirus-vector COVID-19 vaccine is immune thrombotic thrombocytopenia (VITT), which has already been associated with other adenoviral vector vaccines [26]. This condition can be similar to heparin-induced thrombocytopenia (HIT), as both are related to the presence of antibodies against platelet factor 4 (PF4), with thrombocytopenia and consequent thrombosis, although the antigenic target seems to be different in the two situations. In HIT, the heparin binds to PF4 and forms an antigen which triggers IgG autoantibodies, which, in their turn, will cause platelet activation and aggregation, leading to thrombosis. The antibody-covered platelets are then removed from circulation by the endothelial system and spleen, resulting in thrombocytopenia [40]. It has been speculated that in VITT, viral particles of the vaccine would bind to PF4 in a fashion similar to that in patients not previously exposed to heparin. This severe side effect was reported after vaccination with Astra Zeneca (ChAdOx1), Moderna (mRNA-1273) [41] and Johnson & Johnson (Ad26.COV2.S), occurring at a fourfold higher frequency with ChAdOx1 [26]. There have been three other reports of CVT after vaccination with the mRNA-type vaccine by Moderna [42], although patients presented normal platelet counts.

The fact that VITT occurs mainly after vaccination with ChAdOx1 suggests that VITT-induced antibodies against PF-4 do not react to the COVID-19 spike protein but probably react to the adenoviral vector [43].

One German multicentric vaccination study group found an incidence of CVT of 0.55/100,000 in patients after taking vaccines. They reported a total of 45 CVT cases within a month after vaccination, as well as nine ischemic strokes and four hemorrhagic strokes. There was a predominance of females (77.8%) and of patients younger than 60 years (80%); 85.5% of events occurred after ChAdOx1 vaccine, 14.5% after BNT162b2 vaccine; and none after mRNA-1273 vaccine. The adjusted incidence for the ChAdOx1 vaccine was 1.52 per 100,000, with an adjusted incidence of 3.14 for women [4,43,44]. This study pointed towards an increased risk of CVT after the ChAdOx1 vaccine, especially for young women.

One meta-analysis revealed that CVT represented 29% of total thromboembolic events following adenovirus-vector-based vaccination against COVID-19; 28% of patients who had CVT post vaccine died. The mean age of patients who presented CVT post vaccine was 45 years, with 75% of them being women; the mean time from vaccine to symptoms was 10 days, and 68% presented hemorrhagic infarcts [45].

The time of CVT symptoms onset after vaccine ranged from 6 to 15 days after the Ad26.COV2.S vaccine and 5 to 24 days after the ChAdOx1 vaccine [26,46]. The majority of patients did not have any history of thrombosis or coagulation disorders [47]. The outcomes of one case report of 12 patients who developed VITT after Ad26.COV2.S vaccine were 3 deaths, 3 continued ICU care, 2 continued non-ICU hospitalization, and 4 discharged home. Of the 12 patients, 7 developed intracranial hemorrhage (including the 3 individuals who died). The patients presented thrombocytopenia associated with increased D-dimers.

One large cohort study in the UK included 95 patients who presented CVT after COVID-19 vaccines, and it compared the group who fulfilled diagnostic criteria for VITT to the CVT without those criteria. It found that the mean age was lower in the VITT group (47 versus 57). The VITT group had more sinus thrombosed than the non-VITT group, more hemorrhagic infarcts and unfavorable outcomes. Outcomes were better amongst those who received non-heparin anticoagulants and immunoglobulin [48]. Another systematic review found that up to 49% of intracerebral hemorrhage or subarachnoid hemorrhage in CVT was associated with VITT [40]. One study also found VITT to be related to ischemic stroke [43].

There were also two cases of CVT reported after BNT162b2 vaccine, but with no evidence of VITT [49]. One of the patients had anemia and was under hormonal treatment.

A VITT risk score was developed, adapted from the HIT 4Ts scoring system which considers the number of thrombocytes, the timing post-vaccine, the presence of thrombosis or elevated D-dimers and the presence of other possible causes for thrombosis or thrombocytopenia. In the German study, CVTs with a score higher than two occurred after the ChAdOx1 vaccine, and 44% of these CVTs scored four points, meeting all criteria for likelihood of association with the vaccine [43]. The mortality of VITT is high, varying from 18.3% to 50% [26,43].

Clinicians should raise awareness of possible VITT, especially in young female patients presenting with CVT after exposure to COVID-19 vaccines, and they must dose PF4-antibodies if thrombocytopenia, preferentially by ELISA HIT assays (which were shown to have higher sensitivity for detection of these antibodies) [44]. Other techniques are platelet activation assays and serotonin release assays, but they are more influenced by technical aspects and the results are less uniform [26].

Treatment consists of anticoagulation by non-heparin anticoagulants (fondaparinux, argatroban, direct oral anticoagulants) and avoiding heparin and heparin products (including prothrombin complex concentrates) [26]. Warfarin should be avoided, especially in the beginning of treatment, due to its paradoxical induction to a hypercoagulability state. Anticoagulation should be continued for at least three months [26]. Fibrinolysis and mechanical thrombectomy should be restricted to cases with poor evolution despite anticoagulation [17]. Aspirin should also be avoided, due to an increased risk of bleeding, in addition to presenting no benefit.

Intravenous immunoglobulins, at a dose of 1 g/kg for at least 2 days, should be used in refractory cases [46,50]. For patients with a high antibody burden, plasma exchange can be performed in addition to immunoglobulins, for at least 5 days until platelet count improves. There is no consensus regarding the use of corticosteroids.

Platelet transfusions could be beneficial before invasive procedures or for patients with a high risk of bleeding [51]. In HIT, it was related to a fivefold increase in mortality, so it should be avoided [52]. The use of Eculizumab and rituximab are under investigation [26]. There is one case report of treatment of HIT by rituximab with favorable outcomes [53]. Eculizumab was given to two refractory patients who failed IVIG therapy, with improvement [54]. Nevertheless, the risk of CVT is much higher amongst patients infected by COVID-19 as opposed to patients who presented CVT secondary to the vaccine.

The main limitations of this study concern its small number of patients, as well as its retrospective nature. Although we have performed an extensive medical chart review to track patients who presented cerebral venous thrombosis in the context of a COVID-19 infection over the last two years, it is possible that some data are missing. One reason might be faulty medical chart completion. If the study had a prospective nature, medical staff could be made aware of more specific data that should be collected for all possible candidates. It is also possible that we have failed to diagnose COVID-19 in some patients with cerebral venous thrombosis due to fact that in the beginning of the pandemic not all individuals were screened for COVID-19.

## 5. Conclusions

CVT is a rare yet potentially severe complication of COVID-19 infection and also of COVID-19 vaccination. The development of headaches associated with seizures, impaired consciousness or focal signs should raise immediate suspicion of CVT, and neuroimaging should be obtained as soon as possible. In patients who received an adenovirus-vector vaccine and are presenting with thromboembolic events, platelet count, D-dimer and PF4 antibodies should be dosed, and if there is suspicion of VITT, non-heparin anticoagulation should be started in conjunction with immunoglobulins. Nevertheless, since COVID-19 infection itself is also related to thromboembolic events, the benefits of COVID-19 vaccination outweigh the risk of thrombosis and should not be interrupted [43,55]. A special consideration to bear in mind is that adenovirus-vector vaccines were mostly destined for low and middle income countries, since they are more easily stored and require only a single-dose regimen of immunization [8,47,55], so there might be a need to develop comparative ethnographic studies in the future in order to better understand such complex relations.

## Figures and Tables

**Figure 1 life-12-01105-f001:**
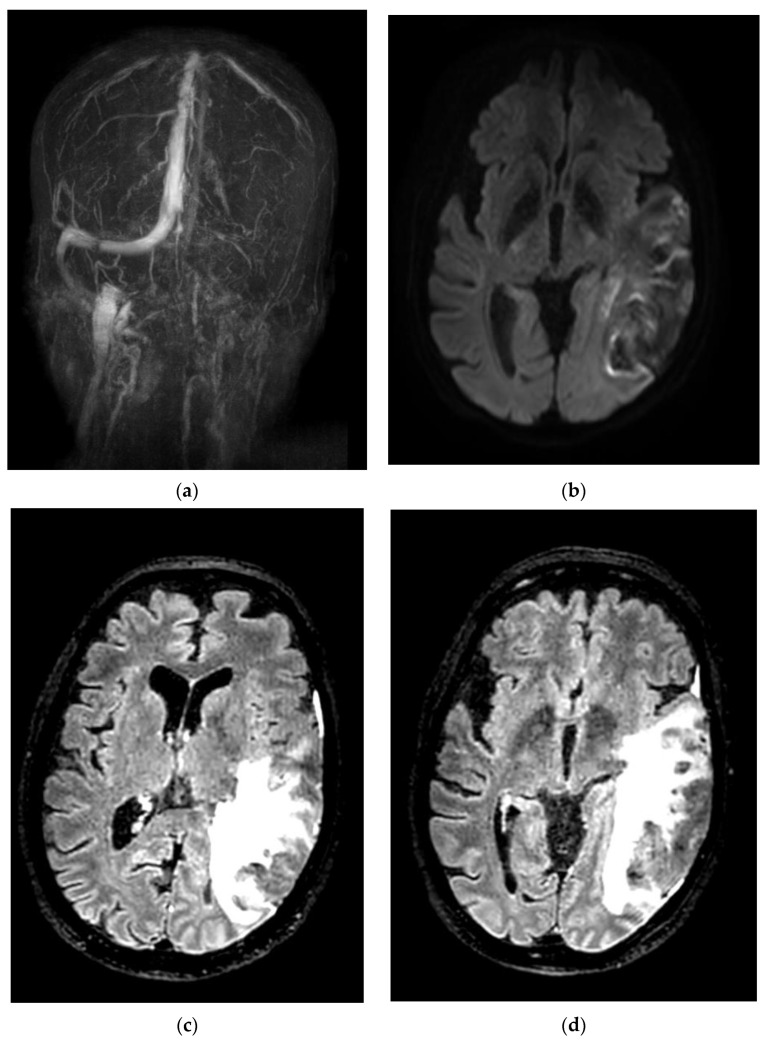
(**a**) Venous MRI showing thrombosis of left transverse and sigmoid sinuses. (**b**) Diffusion-weighted image showing ischemia of left temporal region. (**c**,**d**) Fluid-attenuated inversion recovery (FLAIR) showing hypersignal in temporal and parietal regions.

**Figure 2 life-12-01105-f002:**
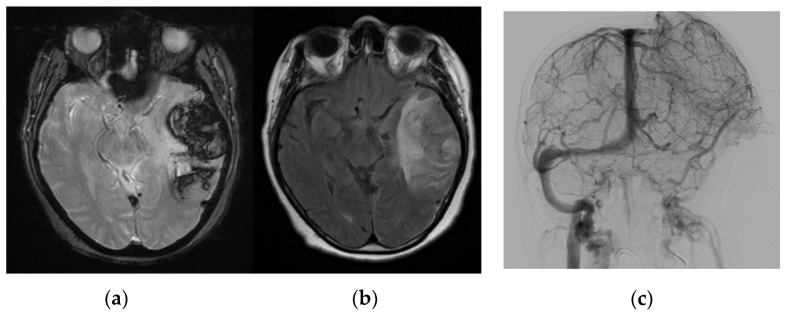
(**a**) Magnetic Resonance Imaging (T2* weighted image). (**b**) FLAIR, showing venous hemorrhagic infarct. (**c**) Venous angiogram showing extended thrombosis of the left transverse and sigmoid sinuses, after craniectomy.

**Figure 3 life-12-01105-f003:**
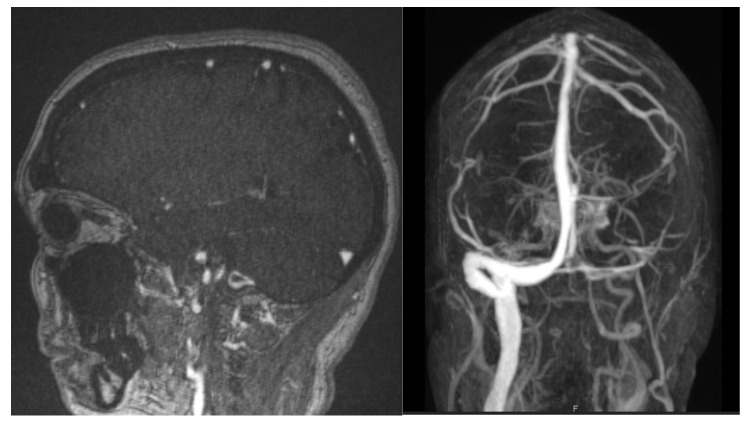
Venous MRI showing thrombosis of left transverse and sigmoid sinuses.

**Table 1 life-12-01105-t001:** Clinical, radiological and epidemiological data of patients presenting CVT and COVID-19.

Gender	Age	Vaccination	Risk Factors	Time from COVID to CVT (Days)	Neuroimage	Treatment	Complications	Disclosure
F	61	No	No	15	Thrombosis in multiple sinuses (internal cerebral vein, Galen vein, right sinus, torcula, left lateral sinus)	LMWH, craniectomy, thrombectomy	Epilepsy	Death
F	54	No	Previous cancer, under Tamoxifen	0	Left transverse and sigmoid sinus thrombosis, with left temporal and parietal hemorrhagic infarct	LMWH	Neuropsychiatric alterations	Transferred to Psychiatry, not fully recovered. Persistence of cognitive disorders.
F	52	Yes (Astra Zeneca)	5 miscarriages, SAPL-negative	18	Right lateral and sigmoid sinus thrombosis	LMWH then apixaban	None	Discharged with no complications
F	31	No	No	30	Right lateral sinus and superior sagittal sinus thrombosis	LMWH	None	Discharged with no complications
F	31	No	Oral contraceptive, anemia Hb 7 g/dL	30	Right lateral and sigmoid sinus thrombosis	LMWH followed by Dabigatran	None	Discharged with no complications

## Data Availability

Data are available upon request.

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
