# Peer review of "Epidemiology and Management of Cerebral Venous Thrombosis during the COVID-19 Pandemic"

_life, 2022, doi:10.3390/life12081105_

Round 1

Reviewer 1 Report

- the English of the paper should be revised due to some typos and sentence construction. Please revise.

- the Introduction section is too long: the authors should reduce the length of this section, move some parts to the discussion section, and be more focused on the main aims of the paper.

- the retrospective nature of this study is a limitation. This should be discussed into a dedicated limitation section. Please provide.

- the small sample size might be considered a limitation of the paper. The authors should discuss such a point in the limitation section.

- Please provide a post-hoc sample size calculation for this paper.

- the authors can consider and discuss the paper from Acanfora D et al. Viruses. 2021 Sep 23;13(10):1904.

Author Response

Dear Reviewer 1,

            We would like to thank you for the helpful insights and pertinent and helpful comments that will increase the quality of this paper.

            Please find below the answer point-by-point to your kind remarks.

  • English has been revised by an English professor with experience in text revision
  • We have shortened the introduction and transferred the content to the discussion
  • A limitation section has been added to address this subject
  • Due to the descriptive design of our case series report and literature review, the analysis was focused on the description of patients (sociodemographic, clinical and imaging characteristics). Continuous variables were described as mean (+/- SD) or median; categorical variables as numbers and percentages. Calculation of post-hoc sample size, therefore, was neither pertinent to our design nor feasible, because 1) the overall incidence of CVT amongst Covid patients is yet not known; 2) only Covid infection in the CVT patients admitted to our unit was tracked, not the incidence of CVT in a sample of Covid patients.
  • We have mentioned and included in the discussion the paper of Alcanfora D. (1).

            Once again thank you for your priceless contribution.

            Best regards,

            Natalia Novaes & Michaël Obadia

  1. Acanfora D, Acanfora C, Ciccone MM, Scicchitano P, Bortone AS, Uguccioni M, et al. The Cross-Talk between Thrombosis and     Inflammatory Storm in Acute and Long-COVID-19: Therapeutic Targets         and Clinical Cases. Viruses. 2021;13(10).

Reviewer 2 Report

Dear Authors,

Congratulation on your very interesting study. The manuscript is very up-to-date however some issues should be raised:

1. All abbreviations should be explained when used first time in  the text, e.g. FLAIR, LMWH.

2. Did the authors get ethical approval to perform a retrospective analysis of medical charts of the number of hospitalizations? Please clarify this.

3. It is not necessary to enter the name of the JASP program as a footnote, please insert it into the text.

4. Along with mean values please show standard deviations.

5. Limitation paragraph should be added at the end of the Discussion section.

Author Response

Dear Reviewer 2,

            We would like to thank you for the pertinent and helpful comments that will increase the quality of this paper. English has been reviewed for this new version.

            Please find below the answer point-by-point to your kind remarks.

  1. Abbreviations have been detailed.
  2. We have obtained oral consent from either patients or their families. This information has been added to the text.
  3. The information about JASP has been added to the text.
  4. Standard deviations were added to the text.
  5. Limitations paragraph was added to the discussion.

Kind regards,

Natalia Novaes & Michaël Obadia